# Significant and Various Effects of ML329-Induced MITF Suppression in the Melanoma Cell Line

**DOI:** 10.3390/cancers16020263

**Published:** 2024-01-07

**Authors:** Nami Nishikiori, Megumi Watanabe, Tatsuya Sato, Masato Furuhashi, Masae Okura, Tokimasa Hida, Hisashi Uhara, Hiroshi Ohguro

**Affiliations:** 1Department of Ophthalmology, Sapporo Medical University, S1W17, Chuo-ku, Spporo 060-8556, Japan; nami076@yahoo.co.jp (N.N.); watanabe@sapmed.ac.jp (M.W.); 2Department of Cardiovascular, Renal and Metabolic Medicine, Sapporo Medical University, S1W17, Chuo-ku, Spporo 060-8556, Japan; satatsu.bear@gmail.com (T.S.); mfuruhas@gmail.com (M.F.); 3Department of Cellular Physiology and Signal Transduction, Sapporo Medical University, S1W17, Chuo-ku, Spporo 060-8556, Japan; 4Department of Dermatology, Sapporo Medical University, S1W17, Chuo-ku, Spporo 060-8556, Japan; haiyingj@sapmed.ac.jp (M.O.); hidat@sapmed.ac.jp (T.H.); uharah@sapmed.ac.jp (H.U.)

**Keywords:** microphthalmia-associated transcription factor (MITF), ML329 (low-molecular MITF-specific inhibitor), malignant melanoma (MM), cell viability, metabolic functions, three-dimensional spheroid culture

## Abstract

**Simple Summary:**

This research explores how the low-molecular microphthalmia-associated transcription factor (MITF) specific inhibitor ML329 affects malignant melanoma (MM) cells’ biology. ML329 significantly reduced cell viability in specific melanoma cell lines and differently influenced their metabolic functions. ML329 also substantially and differently affected the ability to form 3D spheroids and altered the expression of genes related to regulatory and signaling factors among various MM cell lines. These diverse effects across different melanoma types underscore the complexity of MITF-related activities among various types of MM, aiding the development of a more effective understanding of MMs’ pathophysiology as well as their targeted therapies.

**Abstract:**

To study the inhibitory effects on microphthalmia-associated transcription factor (MITF)-related biological aspects in malignant melanomas (MMs) in the presence or absence of the low-molecular MITF specific inhibitor ML329, cell viability, cellular metabolic functions, and three-dimensional (3D) spheroid formation efficacy were compared among MM cell lines including SK-mel-24, A375, dabrafenib- and trametinib-resistant A375 (A375DT), and WM266-4. Upon exposure to 2 or 10 μM of ML329, cell viability was significantly decreased in WM266-4, SK-mel-24, and A375DT cells, but not A375 cells, in a dose-dependent manner, and these toxic effects of ML329 were most evident in WM266-4 cells. Extracellular flux assays conducted using a Seahorse bioanalyzer revealed that treatment with ML329 increased basal respiration, ATP-linked respiration, proton leakage, and non-mitochondrial respiration in WM266-4 cells and decreased glycolytic function in SK-mel-24 cells, whereas there were no marked effects of ML329 on A375 and A375DT cells. A glycolytic stress assay under conditions of high glucose concentrations also demonstrated that the inhibitory effect of ML329 on the glycolytic function of WM266-4 cells was dose-dependent. In addition, ML329 significantly decreased 3D-spheroid-forming ability, though the effects of ML329 were variable among the MM cell lines. Furthermore, the mRNA expression levels of selected genes, including *STAT3* as a possible regulator of 3D spheroid formation, *KRAS* and *SOX2* as oncogenic-signaling-related factors, *PCG1a* as the main regulator of mitochondrial biogenesis, and *HIF1a* as a major hypoxia transcriptional regulator, fluctuated among the MM cell lines, possibly supporting the diverse ML329 effects mentioned above. The findings of diverse ML329 effects on various MM cell lines suggest that MITF-associated biological activities are different among various types of MM.

## 1. Introduction

A malignant melanoma (MM) can arise from the skin and various other organs and tissues including the eyes, gastrointestinal tract, genitalia, and sinuses [1,2,3]. Over the past 10 years, the survival and quality of life of patients receiving treatment for MM have greatly improved due to the development of new MM therapies using various targeted inhibitors for BRAFV600E and MEK kinases as well as immune-based strategies [4,5,6,7]. However, the effectiveness of these targeted and non-targeted immune-based strategies has been poor for MMs lacking the *BRAF* mutation and for rare MM subtypes, including the uveal, acral, and mucosal MM subtypes, due to either primary or acquired resistance following initial treatment [8]. To overcome these problems, additional translational research conducted to develop new therapeutic targets as well as identify new clinical markers is needed. For this purpose, in vitro three-dimensional (3D) culture models in addition to conventional two-dimensional (2D) models have been developed [9]. In our recent studies using a 3D spheroid culture technique, which is a simple 3D cell culture method [10,11,12,13], we successfully established in vitro spheroid models using five different MM cell lines, namely, SK-mel-24, MM418, A375, WM266-4, and SM2-1, as well as dabrafenib- and trametinib-resistant A375 (A375DT) cells. The 3D MM spheroids had non-globe-shaped configurations, as shown in a previous study [14], which contrasted with the globe-shaped 3D spheroids obtained from non-cancerous cells [10,11,12,13]. In addition, interestingly, (1) the degrees of deformity of the 3D spheroids were diverse among the MM cell lines and increased in the following order: WM266-4, SM2-1, A375, MM418, and SK-mel-24. Additionally, (2) the examination of cellular metabolic function using a Seahorse bioanalyzer showed that mitochondrial and glycolytic functions were also different between the less-deformed of the two MM cell lines, WM266-4 and SM2-1, and the more-deformed MM cell lines. RNA sequence analysis of the least- and most-deformed cell lines, WM266-4 and SK-mel-24, respectively, indicated that *KRAS* and *SOX2* were potential master regulatory genes for inducing the differences in biological aspects [14]. Furthermore, the 3D spheroid appearance and cellular metabolic functions of the anti-tumor-drug-resistant A375 and A375DT cells were also different from those of the A375 parent cells [14]. Based on these findings, we suggest that the 3D spheroid configuration is a potential indicator for evaluating the pathophysiological activities of various types of MM cells.

The microphthalmia-associated transcription factor (MITF) gene was initially identified as a gene affecting murine body color, and mice lacking MITF function had a total absence of melanocytes that resulted in a complete lack of pigment in the fur as well as the eyes [15]. MITF is known to be a basic helix–loop–helix leucine zipper (bHLH-ZIP) transcription factor that forms homodimers or heterodimers with other transcription factors, including transcription factor EB (TFEB), transcription factor E3 (TFE3), and transcription factor EC (TFEC), which are members of the microphthalmia-associated transcription factor family (MiT family) that are, unlike MITF, ubiquitously expressed and not essential for melanocytic differentiation [15]. Upon activation, MITF initiates transcriptional events leading to melanocyte differentiation, cell cycle arrest, survival, and the pigmentation of normal melanocytes by regulating the expression of various pigmentation-related genes, including genes encoding tyrosinase [16], tyrosinase-related protein 1 (TYRP-1), dopachrome tautomerase/tyrosinase-related protein-2 (DCT/TYRP-2) [17], silver [18], and absent in melanoma 1 (AIM1) [19]. It has also been suggested that MITF may cause cell cycle arrest during melanocytic differentiation, presumably via the transcriptional targeting of the cyclic-dependent kinase inhibitors p21, CDKN1A, and CDK4A (INK4A) [20]. Alternatively, based on a Bcl-2 knockout resulting in a white body color due to the death of melanocyte cells, Bcl-2, an anti-apoptotic factor, was suggested to be directly involved in MITF activation leading to the survival of melanocytes [21]. In 15% to 20% of MM cases, the amplification and over-expression of MITF leading to bona fide melanoma oncogenes have been found [22]. In fact, a decrease in the 5-year overall survival rate has been shown to be associated with MITF over-expression, and a high MITF expression level is a poor prognostic index in patients. Thus, proliferative and invasive efficacies could be defined by high and low expression levels of MITF [23], and suppression of MITF activity may be an effective therapeutic strategy for MM. As a possible means of inhibiting MITF activity, it has been shown that a small molecule probe called “ML329” can inhibit the expression of numerous MITF target genes and block the proliferation of numerous cell lines requiring MITF for proliferation [24]. A recent study also showed that ML329 significantly suppressed the growth and survival of a metastatic liver uveal melanoma (OMM2.5) [25]. However, at the time of writing this manuscript, suppression of MITF activity via the treatment of MM cells with ML329 has not been sufficiently characterized; therefore, it would be of great interest to investigate the effects of ML329 on our recently developed in vitro 3D MM spheroid models.

To study this issue, in the presence or absence of ML329, various 2D- and 3D-cultured MM cells obtained from three different MM cell lines (SK-mel-24, A375, and WM266-4) and dabrafenib- and trametinib-resistant A375 (A375DT) cells were used for (1) cell viability analysis (2D), (2) real-time cellular metabolic function analysis to evaluate biological activities (2D), analysis of the 3D MM spheroid configuration, and qPCR analysis of possible related genes.

## 2. Materials and Methods

### 2.1. Two-Dimensional Planar and Three-Dimensional Spheroid Cultures of MM Cell Lines

Three human MM cell lines including (1) WM266-4 (CVCL_2765, American Type Culture Collection, Manassas, VA, USA), (2) SK-mel-24 (HTB-71, American Type Culture Collection, Manassas, VA, USA), and (3) A375 (CRL-1619™, American Type Culture Collection, Manassas, VA, USA) were used in this study. The A375DT (A375- dabrafenib-/trametinib-resistant) cell line established from A375 cells by selecting monoclones using the limiting dilution method after treatment with gradually increasing concentrations of dabrafenib and trametinib (as described previously [26]) was also used in this study. These 4 cell lines were 2D- and 3D-cultured using methods described in our previous reports [27] (information on methods used in this study is shown in Appendix A). On Day 7, both the 2D- and 3D-cultured cells were collected and further processed for use in the analyses described below.

### 2.2. Assessment of Mitochondrial and Glycolytic Functions of Various MM Cell Lines

The oxygen consumption rate (OCR) and extracellular acidification rate (ECAR) of the 2D-cultured MM cell lines, including WM266-4, SK-mel-24, A375, and A375DT cells, that were untreated (control) or treated with 2 or 10 μM of ML329 for 24 h were determined using a Seahorse XFe96 real-time extracellular flux analyzer (Agilent Technologies, Santa Clara, CA, USA) according to the manufacturer’s instructions (information on methods used in this study is shown in Appendix A).

### 2.3. Phase Contrast Microscopy of 3D Spheroids Derived from Various Human MM Cell Lines

The morphology of the 3D MM spheroids was observed using a phase contrast microscope (Nikon ECLIPSE TS2; Tokyo, Japan) and a micro-monitoring camera equipped with a microsqueezer (MicroSquisher, CellScale, Waterloo, ON, Canada) as described previously [10].

### 2.4. Other Analytical Methods

Cell viability assays for the 2D cultures of MM cell lines were called out using a Cell Counting Kit-8 (Dojindo, Tokyo, Japan) as described in the protocol. Total RNA extraction, reverse transcription, real-time PCR, and quantification of the respective genes using specific primers (Appendix A) were described in a previous report [13]. All statistical analyses were performed using Graph Pad Prism 8 (GraphPad Software, San Diego, CA, USA). The statistical difference between groups was determined using Student’s *t*-test for two-group comparison or one-way or two-way ANOVA followed by Tukey’s multiple comparison test depending on the compared groups. Data are expressed as arithmetic means ± the standard error of the mean (SEM).

## 3. Results

To elucidate the possible roles of MITF in the biological aspects of various types of MM, a specific MITF inhibitor, ML329, and recently established 3D spheroids derived from WM266-4, A375, A375DT, and SK-mel-24 cells, in which the morphology and biological functions were distinct, were used [27]. Among the four different MM cell lines, (1) the deformity of the horizontal 3D spheroid configuration was increased, and the mitochondrial respiratory and glycolytic capacities were decreased and increased, respectively, in the following order: WM266-4, A375, and SK-mel-24; additionally, (2) A375DT cells showed a different horizontal 3D spheroid configuration as well as different mitochondrial respiration and glycolytic capacity from those of the other cells [27].

Initially, to study the effects of ML329 on the cell viability of the four MM cell lines, in which gene expressions of MITF were detected but wherein the levels of which were different among the different lines (Appendix A), 2 or 10 μM concentrations were used, as reported in a previous study [24] and according to the recommendation by the company that produced the product. As shown in Figure 1, cell viability was significantly decreased in WM266-4, SK-mel-24, and A375DT cells but not in A375 cells in the presence of ML329 in a dose-dependent manner, and the ML329-induced changes were most notable in WM266-4 cells.

Similar to the different effects of ML329 on cell viability, the cellular metabolic functions evaluated using a Seahorse bioanalyzer were also different among the MM cell lines, as shown in Figure 2. In the MW266-4 cells, treatment with ML329 significantly increased the basal OCR (i.e., basal respiration), ATP-linked respiration, proton leakage, and non-mitochondrial respiration, while spare respiratory capacity was decreased (Figure 2C). The effect of ML329 on glycolytic functions in MW266-4 cells was slight, regardless of the dose (Figure 2D,E). A375 and A375DT cells did not show marked effects of ML329 on cellular metabolism (Figure 2F–N). However, in SK-mel-24 cells, treatment with ML329 induced mild but significant increases in basal respiration, ATP-linked respiration, and non-mitochondrial respiration, while a marked decrease in glycolytic function was observed (Figure 2O–S).

To further investigate the effect of ML329 on glycolytic function in SK-mel-24 cells, glycolytic stress tests were performed at different glucose concentrations. As shown in Figure 3, ML329 inhibited both glycolysis and glycolytic capacity at low (5.5 mM) and high (11 mM) glucose concentrations, and significant dose-dependent effects of ML329 on glycolysis and glycolytic capacity were observed under a high-glucose concentration condition. These findings suggest that suppression of MITF activity by ML329 has different effects on cellular viability and metabolic functions in different melanoma cells, and these effects may be prominent in WM226-4 and SK-mel-24 cells.

Next, the effects of ML329 on 3D spheroid morphology were determined. As shown in Figure 4, the capacity for the formation of 3D spheroids was significantly decreased and resulted in sparse 3D spheroids in A375 cells in the presence of 2 μM and 10 μM of ML329 and in A375DT cells and SK-mel-24 cells in the presence of 10 μM ML329, and the capacity for 3D spheroid formation in WM266-4 cell was almost lost, even in the presence of only 2 μM of ML329.

To study these issues further, we evaluated the gene expression of the following important factors: (1) *STAT3*, a possible master regulator for 3D spheroid formation [28]; (2) *KRAS* and *SOX2*, possible regulators of the induction of deformity in MM cell lines [27]; (3) *PGC1a*, the main regulator of mitochondrial biogenesis [29]; and (4) *HIF1a*, a major hypoxia transcriptional regulator [30], among A357, WM266-4, and Skmel-24 cells (Figure 5). Upon exposure to ML329, the mRNA expression of *STAT3*, *KRAS*, and *SOX2* was significantly up-regulated and that of *PGC1a* and *HIF1a* was significantly down-regulated in WM266-4 cells. In contrast, significant up-regulation of *HIF1a* and *PGC1a* and substantial down-regulation of *SOX2* were observed in A357 cells and Skmel-24 cells, respectively. Taken together, these results suggest that ML329 has effects on cellular biological functions and that the 3D spheroid formation abilities of the three different MM cell lines may be caused by different mechanisms. 

The results of this study suggest that the inhibition of MITF activity by ML329 had significant effects not only on the 3D architectures of MM cell lines but also on their biological aspects, and these effects were unique depending on the different MM cell origins.

## 4. Discussion

In agreement with the MITF expression levels, two main groups of MM cells have been identified: a high-MITF-expression (MITF-high) group with a high proliferation rate and low invasive potential and a low-MITF-expression (MITF-low) group with a low proliferation rate and high invasive potential [31]. Since both MITF-low cells and MITF-high cells can be mixed within tumors [32,33], it is thought that the phenotypes of these two groups of cells can be switched to fluctuate their proliferative or invasive efficacies without genetic alteration [32,34]. Although the precise molecular mechanisms by which such phenotype switching is induced in the two groups of cells are not clear, hypoxia and nutrient starvation have been suggested to be possible factors involved [32,35]. In fact, MITF regulates the expression of the metabolic factor *PGC1a* [36,37], and MITF binds to the *HIF1a* promoter to stimulate its transcriptional activity in MM cells [38]. In the current study, cellular metabolic states determined using a Seahorse bioanalyzer and phenotypes determined according to 3D spheroid forming efficacy were significantly modulated by a specific inhibitor of MITF activity in a concentration-dependent manner, and they were also different among the MM cell lines A375, WM266-4, and Skmel-24 as well as between the parent A375 cells and the anti-tumor drug-resistant A375DT cells. The concentration-dependent inhibitory effect of the MITF inhibitor ML329 on glycolytic function in SK-mel-24 cells under high-glucose-concentration conditions observed in the present study clearly indicates that the activation of MITF is associated with the HIF-1α-mediated enhancement of glycolytic activity in some melanoma cells.

It has been shown that MITF can promote the survival of MM cells through several anti-apoptotic mechanisms [39,40,41,42]. However, as of this writing manuscript, although no direct MITF inhibitors are available, several studies related to the potential upstream regulators and/or downstream effectors of MITF have revealed the following therapeutic candidates: (1) inhibitors of HDAC (histone deacetylase), which decreased MITF expression and inhibited tumor growth in a human cutaneous melanoma xenograft model [43]; (2) the HDAC trichostatin A (TSA), which increased the expression of miR-137, which, in turn, reduced MITF expression in uveal melanoma cells [44]; (3) a small molecule inhibitor of p300/CBP, which dramatically reduced the number of human cutaneous melanoma cells in vitro [45]; and (4) both the HDAC inhibitor ACY-1215 and a small molecule inhibitor of the MITF pathway (ML329), which reduced the proliferation of a metastatic uveal melanoma (UM) cell line in vitro [25]. In the current study, ML329 exerted toxic effects, but these effects were variable among the MM cell lines evaluated. Interestingly, the toxic effects of ML329 on WM266-4 cells were significantly enhanced to a lethal level in the presence of glucose in a concentration-dependent manner. In support of the observation in our study, a previous study showed that the MITF expression levels in several MM cell lines, including 501mel, A375, and WM266-4 cells, were significantly increased with an increase in the glucose concentration in the culture medium, suggesting that the demand for MITF is increased by an increase in glucose concentration [46]. Furthermore, a recent study showed that CCDC183-AS1 is involved in bladder cancer (BC) progression via an MITF/CCDC183-AS1/miR-4731-5p/TCF7L2 signaling axis [47], in which TCF7L2 was identified as an essential gene in the modulation of aerobic glycolysis (the Warburg effect) [48,49,50]. Based on these results, we reasonably concluded that the suppression of MITF activity by ML329 greatly affects the viability of MM cell lines by modulating cellular metabolic functions related to glycolysis as well as mitochondrial functions.

However, as a study limitation, this conclusion remains speculative in terms of the diverse effects of ML329 induced in various MM cell lines as well as the positive and negative states of anti-tumor-drug sensitivity because of insufficient evidence of the up-stream and down-stream mechanisms responsible. Thus, to obtain a better understanding of the ML329-induced MITF inhibitory effects in MM, additional investigations with the objective of identifying additional unidentified factors will be our next project.

## 5. Conclusions

In conclusion, our study supports the existence of distinct MM cell types based on MITF expression levels based upon various characteristics without genetic alterations, that is, MITF’s role in (1) cell viability; (2) regulating cellular metabolic functions, particularly glycolytic activity potentially mediated by HIF-1α; and (3) 3D spheroid-forming efficacies. However, despite valuable insights, limitations exist, particularly regarding the variability in ML329’s effects and the incomplete understanding of the involved mechanisms among such distinct MM cell types. Further investigations targeting unidentified factors will enhance our comprehension of the pathophysiological roles of MITF in various MM cells.

## Figures and Tables

**Figure 1 cancers-16-00263-f001:**
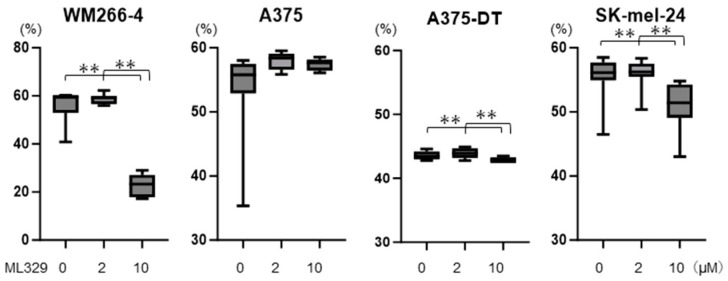
Cell viability of MM cell lines. WM266-4, A375, A375DT, or SK-mel-24 cells were untreated or treated with 2 or 10 μM of ML329 for 24 h, and their viabilities were determined using a commercially available CCK-8 assay kit; these are plotted later on. All experiments were performed in triplicate, with a freshly prepared 2D culture (n = 3) used in each experimental condition. ** *p* < 0.01.

**Figure 2 cancers-16-00263-f002:**
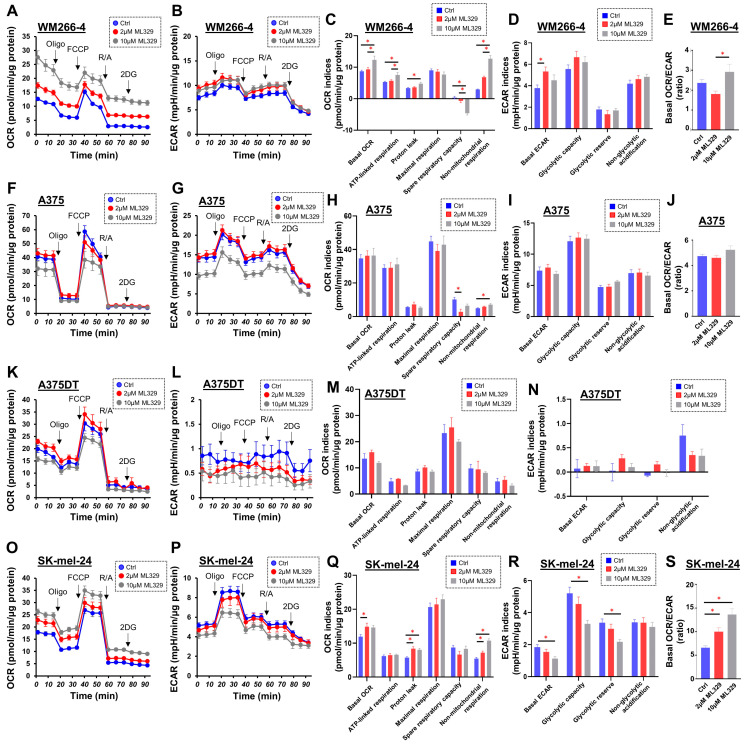
Assessment of the mitochondrial and glycolytic functions of four MM cell lines. The mitochondrial and glycolytic functions of the four different 2D-cultured MM cell lines, namely, WM266-4 (Panels (**A**–**E**)), A375 (Panels (**F**–**J**)), A375DT (Panels (**K**–**N**)), and SK-mel-24 (Panels (**O**–**S**)), assessed, which were untreated (Ctrl) or treated with 2 μL or 10 μL of ML329, were measured using a Seahorse XFe96 bioanalyzer. Key parameters of mitochondrial and glycolytic functions were determined as follows: Basal OCR: OCR (Baseline)−OCR (R/A), ATP-linked respiration: OCR (Baseline)−OCR (Oligo), Proton leak: OCR (Oligo)−OCR (R/A), Maximal respiration: OCR (FCCP)−OCR (R/A), Spare respiratory capacity: OCR (FCCP)−OCR (Baseline), Non-mitochondrial respiration: OCR (Baseline), Basal ECAR: ECAR (Baseline)−ECAR (2DG), Glycolytic capacity: ECAR (Oligo)−ECAR (2DG), ECAR (Oligo)–ECAR (Baseline), Non-glycolytic acidification: ECAR (2DG), and Basal OCR/ECAR: ratio of Basal OCR to Basal ECAR. The Basal OCR/ECAR ratio for A375DT cells is not presented because the values of ECAR in A375DT cells were extremely low and highly variable. All data are represented as means ± standard error of the mean (SEM). All experiments were performed using fresh preparations (n = 5–8). Oligo: oligomycin (complex V inhibitor); FCCP: carbonyl cyanide p-trifluoromethoxyphenylhydrazone (protonphore); R/A: a mixture of rotenone/antimycin A (complex I/III inhibitors); 2DG: 2-deoxylglucose (hexokinase inhibitor). * indicates a *p* value < 0.05.

**Figure 3 cancers-16-00263-f003:**
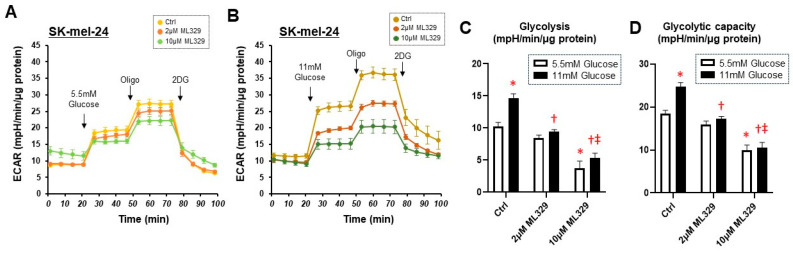
Glycolytic stress tests in SK-mel-24 cells with different glucose concentrations. The glycolytic functions of SK-mel-24 were determined using a glycolytic stress test protocol under the condition of 5.5 mM (Panel (**A**)) or 11 mM (Panel (**B**)) glucose injections using a Seahorse XFe96 bioanalyzer. Glycolysis defined as ECAR (Glucose)−ECAR (Baseline) (Panel (**C**)) and Glycolytic capacity defined as ECAR (Oligo)−ECAR (Baseline) (Panel (**D**)) are represented. All experiments were performed using fresh preparations (n = 4). Oligo: oligomycin (complex V inhibitor); FCCP: carbonyl cyanide p-trifluoromethoxyphenylhydrazone (protonphore); R/A: a mixture of rotenone/antimycin A (complex I/III inhibitors); 2DG: 2-deoxylglucose (hexokinase inhibitor). * indicates a *p* value < 0.05 vs. Ctrl Glucose 5.5 mM, † indicates a *p* value < 0.05 vs. Ctrl Glucose 11 mM, ‡ indicates a *p* value < 0.05 vs. Ctrl Glucose 11 mM.

**Figure 4 cancers-16-00263-f004:**
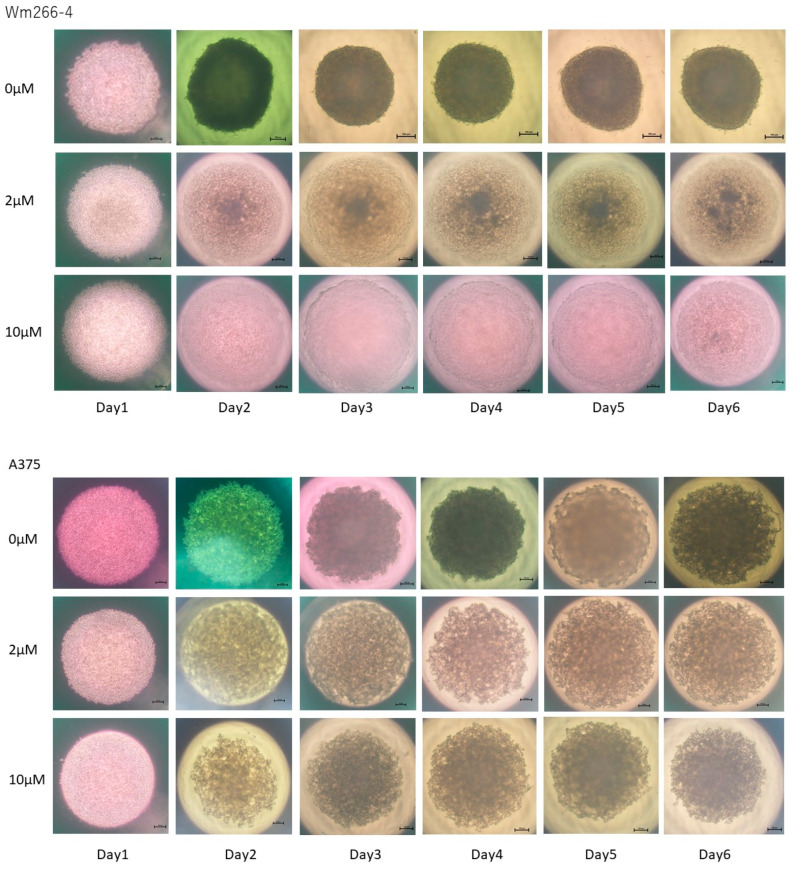
Effects of ML329 on the appearance of 3D spheroids obtained from four different MM cell lines. WM266-4, A375, A375DT, or SK-mel-24 cells were untreated or treated with 2 or 10 μM of ML329 for 24 h, and their 3D spheroid formation was evaluated using phase contrast microscopy. Representative PC images are shown. Scale bar: 100 μm.

**Figure 5 cancers-16-00263-f005:**
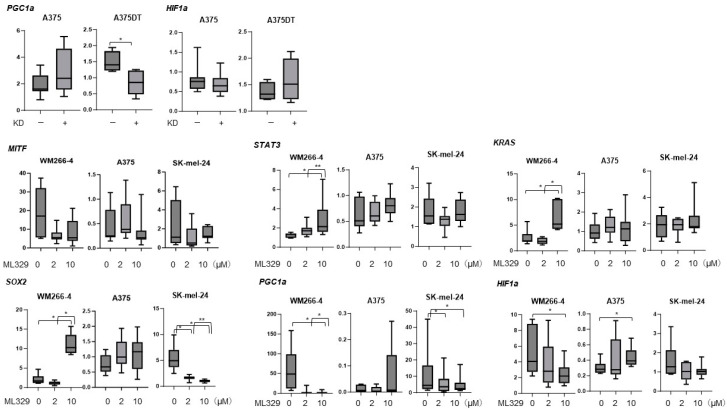
Comparison of the mRNA expression levels of selected genes related to MM pathology. WM266-4, A375, A375DT, or SK-mel-24 cells were subjected to different treatments: untreated, treated with 2 or 10 μM of ML329 for 24 hours, or subjected to MITF gene knockdown (KD), and the mRNA expression levels of *KRAS*, *SOX2*, *MITF*, *STAT3*, *PGC1a,* and *HIF1a* were evaluated using a qPCR procedure. All experiments were performed in triplicate, with a freshly prepared 2D culture (n = 3) used in each experimental condition. * *p* < 0.05; ** *p* < 0.01.

## Data Availability

The data can be shared on request.

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
