# Peer review of "Significant and Various Effects of ML329-Induced MITF Suppression in the Melanoma Cell Line"

_cancers, 2024, doi:10.3390/cancers16020263_

Round 1
Reviewer 1 Report
Comments and Suggestions for Authors
- Figure1
- Treatment of cells with ML329 has not been described in the methods or supplemental methods. How long is this treatment performed for?
- CCK-8 assay- how long are the cells treated with ML329 before the assay was performed?
- Please correct the typo in figure 1 legend in line 171.
- Figure 2
- panels- A-B, F-G, K-L, P-Q- X axis is illegible.
- all panels-legends on top right are unclear.
- Lines 182-`86 the reference of the figure panels in the text is misleading. Panels K-J show SK-mel-24 cells but text describes these as panels O-S. The figure legend needs to rectified too.
- No statistical significan noted for any of the #D cultures as compared to control
- Figure 5
- Figure panel markings missing.
- After marking the figure panels the same should be indicated in the text in lines 250-254.
- Protein expression of these markers needs to be evaluated
General questions-
- What is the expression level of MITF in these cell lines/3D cultures?
- What is the expression of MITF after ML329 treatment in these cultures?
- Why are these particular concentrations of ML329 used? Please cite previous literature if any.
Author Response
Dear Editor,
Thank you very much for the constructive comments concerning our manuscript, " Significant and various effects by ML329 induced MITF suppression in the melanoma cell lines”. We carefully checked all of the Editor and Reviewer comments and prepared a revised version of our paper that takes these comments into account. The changes are listed below.
Reviewer 1
- Figure1
- Treatment of cells with ML329 has not been described in the methods or supplemental methods. How long is this treatment performed for?
Answer; Thank you for this comment. We performed 24 hours exposure of ML329, and thus this information is included in the method; “The oxygen consumption rate (OCR) and extracellular acidification rate (ECAR) of the 2D cultured MM cell lines including WM266-4, SK-mel-24, A375 and A357DT cells that were treated or not treated with control 2 μM or 10 μM of ML329 for 24 hours were determined by using a Seahorse XFe96 real-time extracellular flux analyzer (Agilent Technologies) according to the manufacturer’s instructions (Information on methods used in this study is shown in Supplemental Methods.).”, Fig. 4 legend; “WM266-4, A375, A375DT or SK-mel-24 cells were untreated or treated with 2 or 10 mM ML329 for 24 hours, their 3D spheroid formation were evaluated by phase contrast microscopy. Representative PC images are shown. Scale bar: 100 mm.”, and Fig. 5 legend; “WM266-4, A375, A375DT or SK-mel-24 cells were untreated or treated with 2 or 10 mM ML329 for 24 hours, the mRNA expression levels of KRAS, SOX2, MITF, STAT3, PGC1a and HIF1a were evaluated by a qPCR procedure. All experiments were performed in triplicate with a freshly prepared 2D culture (n=3) used in each experimental condition. *P<0.05, **P<0.01.”.
- CCK-8 assay- how long are the cells treated with ML329 before the assay was performed?
Answer; Thank you for this comment. We performed 24 hours exposure of ML329, as above, and thus this information is included in the Fig. 1 legend; “WM266-4, A375, A375DT or SK-mel-24 cells were untreated or treated with 2 or 10 mM ML329 for 24 hours, their viabilities were determined using a commercially available CCK-8 assay kit, and they are plotted in the future. All experiments were performed in triplicate with a freshly prepared 2D culture (n=3) used in each experimental condition. **P<0.01.”.
- Please correct the typo in figure 1 legend in line 171.
Answer; Thank you for this comment. As pointed out, typos in Figure 1 were properly corrected.
- Figure 2
- panels- A-B, F-G, K-L, P-Q- X axis is illegible.
- all panels-legends on top right are unclear.
- Lines 182-`86 the reference of the figure panels in the text is misleading. Panels K-J show SK-mel-24 cells but text describes these as panels O-S. The figure legend needs to rectified too.
- No statistical significance noted for any of the 2D cultures as compared to control
Answer; Thank you very much for pointing out these errors. The problems seem to have been mainly caused by a garbled PowerPoint file. In the revised version, we have attached Figure 2 as a high-resolution TIFF file to clarify the X-axis and the legend. We have also amended the legend and panel discrepancies, and corrected the statistics to make them clearer.
- Figure 5
- Figure panel markings missing.
Answer; Thank you for this comment. As pointed out, Figure panel markings are included.
- After marking the figure panels the same should be indicated in the text in lines 250-254.
Answer; Thank you for this comment. As pointed out, corresponding text are properly corrected.
- Protein expression of these markers needs to be evaluated
Answer; Thank you for this comment. In terms of the protein-levels analysis of the 3D spheroid, it is very difficult because total protein amounts within a 3D spheroid are extremely small and thus are insufficient for those analysis. However, in our previous studies using various source of cells such as adipocyte, fibrocytes, as well as several cancerous cells, immune-staining intensities using various specific antibodies including ECM proteins, MMPs TIMPs and others were almost comparable levels with gene expression levels. Therefore, we assume that at least, differences among cell lines within the Fig. 5 could be within an allowance. Even though there are some difference between gene expression levels and protein expression levels, since we want show the difference among experimental conditions with or without ML329, our current data of Fig. 5 should be still informative.
General questions-
- What is the expression level of MITF in these cell lines/3D cultures?
Answer; Thank you for this comment. As suggested MITF expression levels were included in the second paragraph of result; “Initially, to study the effects of ML329 on cell viability of the four MM cell lines in which gene expressions of MITF were detected but those levels were different among these (supplemental Fig. 1), 2 or 10 mM concentrations were used as according to the previous literature [24] and the recommendation by the product company.”. Dr. Nishikiori, please check.
- What is the expression of MITF after ML329 treatment in these cultures?
Answer; Thank you for this comment. As shown in Fig. 5, the MITF expression levels were not significantly modulated.
- Why are these particular concentrations of ML329 used? Please cite previous literature if any.
Answer; Thank you for this comment. In terms of the ML329 concentrations, those were followed previous studies related malignant melanoma cells in addition to product maker recommendation. Those information were included in the second paragraph of result; “Initially, to study the effects of ML329 on cell viability of the four MM cell lines, 2 or 10 mM concentrations were used as according to the previous literature [28] and the recommendation by the product company.”.
Reviewer 2 Report
Comments and Suggestions for Authors
Manuscript by Nishikiori et al. includes a description of research conducted on malignant melanoma cell lines in an in vitro model. The tests were performed on a 2D and 3D in vitro model using the small-molecule specific inhibitor MITF-ML329. The authors performed cytotoxicity tests of two concentrations of ML329, assessed cellular metabolism under the influence of this compound, and also conducted a broad analysis of gene expression.
After analyzing the databases, there are few reports on the anticancer effect of ML329, therefore the research conducted by the authors seems innovative and new.
The references used by the authors are consistent with the topic, but most of them are older than 10 years, so the reviewer's suggestion would be to find more current scientific publications.
Regarding the research conducted by the authors, the reviewer has several questions and comments that require clarification and supplementation.
1. Why were only concentrations 2 and 10 of ML329 used in cytotoxicity studies? Have any studies been carried out on the toxicity of this compound on healthy cells, e.g. melanocytes?
2. In the reviewer's opinion, it is necessary to present the results in Figure 1 differently. Unfortunately, they are quite incomprehensible. It was difficult for the reviewer to estimate which of the lines were most and least sensitive to the compound used based solely on the graphs. Moreover, there are no descriptions on the axes regarding the examined parameter and units.
3. Unfortunately, Figure 2 also needs improvement. Due to the number of graphs in one Figure, they are unfortunately completely illegible, even after close zooming. In the charts in subsections A,B,F,G,K,L,P,Q, the numbers on the x-axis are completely blurred. The captions on the remaining charts in this Figure are also difficult to read, especially on the x-axis.
4. Assessing gene expression at the mRNA level provides information about the amount of specific transcripts present in a cell, but it doesn't directly reflect the actual levels of functional proteins. It is essential to perform protein-level analyses, such as Western blotting, to gain insights into the functional and regulatory aspects of gene expression, including post-translational modifications, protein stability, alternative splicing, and cellular localization.
Could the authors explain why they did not perform tests at the protein level?
Best regards.
Comments on the Quality of English LanguageMinor editing of English language required.
Author Response
Dear Editor,
Thank you very much for the constructive comments concerning our manuscript, " Significant and various effects by ML329 induced MITF suppression in the melanoma cell lines”. We carefully checked all of the Editor and Reviewer comments and prepared a revised version of our paper that takes these comments into account. The changes are listed below.
Reviewer 2
Manuscript by Nishikiori et al. includes a description of research conducted on malignant melanoma cell lines in an in vitro model. The tests were performed on a 2D and 3D in vitro model using the small-molecule specific inhibitor MITF-ML329. The authors performed cytotoxicity tests of two concentrations of ML329, assessed cellular metabolism under the influence of this compound, and also conducted a broad analysis of gene expression.
After analyzing the databases, there are few reports on the anticancer effect of ML329, therefore the research conducted by the authors seems innovative and new.
The references used by the authors are consistent with the topic, but most of them are older than 10 years, so the reviewer's suggestion would be to find more current scientific publications.
Answer; Thank you for this comment, As suggested, references (#15-22) were updated.
Regarding the research conducted by the authors, the reviewer has several questions and comments that require clarification and supplementation.
- Why were only concentrations 2 and 10 of ML329 used in cytotoxicity studies? Have any studies been carried out on the toxicity of this compound on healthy cells, e.g. melanocytes?
Answer; Thank you for this comment. The cytotoxic effects toward various cells including healthy cells in addition to malignant melanoma cells were carefully determined, and thus those information were included; “Initially, to study the effects of ML329 on cell viability of the four MM cell lines, 2 or 10 mM concentrations were used as according to the previous literature [24] and the recommendation by the product company.”.
- In the reviewer's opinion, it is necessary to present the results in Figure 1 differently. Unfortunately, they are quite incomprehensible. It was difficult for the reviewer to estimate which of the lines were most and least sensitive to the compound used based solely on the graphs. Moreover, there are no descriptions on the axes regarding the examined parameter and units.
Answer; Thank you for this comment. As suggested, to compare more easily among cell lines, Figure 1 was changed by using percent cell viabilities.
- Unfortunately, Figure 2 also needs improvement. Due to the number of graphs in one Figure, they are unfortunately completely illegible, even after close zooming. In the charts in subsections A,B,F,G,K,L,P,Q, the numbers on the x-axis are completely blurred. The captions on the remaining charts in this Figure are also difficult to read, especially on the x-axis.
Answer; We sincerely appreciate your important suggestions. In the revised version, we have attached revised Figure 2 as a high-resolution TIFF file to clarify the images. We have also amended the legend and panel discrepancies.
- Assessing gene expression at the mRNA level provides information about the amount of specific transcripts present in a cell, but it doesn't directly reflect the actual levels of functional proteins. It is essential to perform protein-level analyses, such as Western blotting, to gain insights into the functional and regulatory aspects of gene expression, including post-translational modifications, protein stability, alternative splicing, and cellular localization. Could the authors explain why they did not perform tests at the protein level?
Answer; Thank you for this comment. In terms of the protein-levels analysis of the 3D spheroid, it is very difficult because total protein amounts within a 3D spheroid are extremely small and thus are insufficient for those analysis. However, in our previous studies using various source of cells such as adipocyte, fibrocytes, as well as several cancerous cells, immune-staining intensities using various specific antibodies including ECM proteins, MMPs TIMPs and others were almost comparable levels with gene expression levels. Therefore, we assume that at least, differences among cell lines within the Fig. 5 could be within an allowance. Even though there are some difference between gene expression levels and protein expression levels, since we want show the difference among experimental conditions with or without ML329, our current data of Fig. 5 should be still informative.
Round 2
Reviewer 1 Report
Comments and Suggestions for Authors
The authors have addressed the comments and concerns sufficiently. i recommend the manuscript for publication.
Reviewer 2 Report
Comments and Suggestions for Authors
The reviewer accepts the authors' arguments and recommends the manuscript for publication.